# Reports of my demise are greatly exaggerated:
## $N$-subjettiness taggers take on jet images

**Liam Moore[1,⋆], Karl Nordström[2,3], Sreedevi Varma[4] and Malcolm Fairbairn[4]**

**1** Center for Cosmology, Particle Physics and Phenomenology (CP3), Université Catholique
de Louvain, Chemin du Cyclotron 2, B-1348 Louvain-la-Neuve, Belgium
**2** National Institute for Subatomic Physics (NIKHEF)
Science Park 105, 1098 XG Amsterdam, Netherlands
**3** Laboratoire de Physique Theorique et Hautes Energies (LPTHE),
UMR 7589 CNRS & Sorbonne Université, 4 Place Jussieu, F-75252, Paris, France
**4** Theoretical Particle Physics and Cosmology, Physics, King's College London,
London WC2R 2LS, United Kingdom

⋆ liam.moore@uclouvain.be

## Abstract

We compare the performance of a convolutional neural network (CNN) trained on jet images with dense neural networks (DNNs) trained on $n$-subjettiness variables to study the distinguishing power of these two separate techniques applied to top quark decays. We find that they perform almost identically and are highly correlated once jet mass information is included, which suggests they are accessing the same underlying information which can be intuitively understood as being contained in 4-, 5-, 6-, and 8-body kinematic phase spaces depending on the sample. This suggests both of these methods are highly useful for heavy object tagging and provides a tentative answer to the question of what the image network is actually learning.


# 1 Introduction

New particles created above the electroweak scale are often expected to decay into the most massive members of the Standard Model family, namely the $W$ and $Z$ bosons, the (Brout-Englert-)Higgs boson $h$ and the top quark. Distinguishing between different heavy particles is challenging if they subsequently decay into lighter coloured particles since these form jets as the initial partons radiate a 'shower' of hadrons. At a hadron collider such as the LHC, the multiplicity of jets which can be produced in a single collision by simple QCD processes is very large, and pile-up and minimum bias hadronic activity further complicate the final state. It can therefore be extremely difficult to map hadronic activity in the final state to the hadronic decay of a heavy particle. This makes it difficult to separate signal from backgrounds in many searches for new physics, and is the dominant issue for measurements of some important Standard Model predictions such as the $h \to b\bar{b}$ branching ratio [1].

The situation can be improved if one considers the substructure of jets which contain within them all of the decay products of a boosted heavy object. In this case the inherent scales inside the jets allow them to be separated from largely scaleless light quark and gluon jets, alleviating the combinatorial backgrounds which make an analysis using fully resolved decay products challenging. Following the early work of Seymour [2], the most successful applications have been at the LHC where for example the BDRS method [3], John Hopkins tagger [4], shower deconstruction [5,6], and HEPTopTagger [7–9] all have seen use by the experiments as well as considerable phenomenological attention.

As the LHC approaches its high luminosity runs, it is also necessary to consider ways to look for new physics in as many directions as possible to make sure we make the most of the available data. It is therefore additionally clear that efficient algorithms for identifying heavy particle decays which can handle large data volumes and don't require hand-tuning for specific decays will become increasingly important in the future, which suggests use of machine learning techniques as an obvious avenue to consider.

One of the most widely studied problems in the larger machine learning community is the task of extracting classification information from large image datasets. It is therefore natural to consider the possibility of converting information from LHC events into images on which we can train a machine learning algorithm on in order to teach it the mapping from hadronic activity to various classifications such as heavy particle decay and 'QCD background' or light quark and gluon jets [10–12]. These kind of techniques have seen attention for example in quark/gluon classification [13], top-tagging [14,15] and $W$ tagging [10].

There have also been initial studies of the sensitivity of these algorithms to the modeling uncertainties involved in the use of Monte Carlo event generators which (just to mention one issue among many) rely entirely on phenomenological models to hadronise the final state after the (well-understood) parton shower has increased the coloured particle multiplicity considerably [16]. Such questions of the extent to which we might be teaching the machine spurious modeling details rather real physics become more pertinent as more advanced algorithms such as jet images squeeze more information out of the radiation patterns. It is therefore interesting to compare the performance of jet image networks to ones with a more intuitive physical interpretation.[1]

$N$-subjettiness variables were first discussed in [17] as a jet substructure application of the ideas introduced by $n$-jettiness [18]. These quantify how much the radiation contained within a jet (event) is aligned along different (sub)jet axes within it. By analysing the bunching and spatial distribution of jets and subjets in an event using this procedure, one can powerfully dis-

---

[1]The usual reply is of course that these algorithms should ultimately be trained on purified data samples and not simulation, but a good understanding of the physics they are learning is still necessary and identifying issues with the modeling could also potentially help the Monte Carlo generator development community.

tinguish between different decay and event topologies. The $n$-subjettiness variables were first applied to distinguishing $W$ bosons and top quarks from QCD backgrounds in [17, 19], and have subsequently seen wide use in the phenomenological literature and by the ATLAS and CMS experiments (see, e.g. [20–22]). Later on similar ideas led to the development of Energy Correlation Functions [23, 24] and most recently Energy Flow Polynomials (EFPs) [25]. The $n$-subjettiness variables can also be used as inputs for a machine learning algorithm and this approach can perform very well at heavy object discrimination [25–27]. Such an approach is additionally attractive since the small number of input variables here allows for a mapping to the $m$-body kinematic phase space of the jet constituents, which hints at an intuitive understanding of what the machine is actually learning.

Indeed, when applying such techniques to jet tagging, it would be very nice to figure out precisely "What is the machine learning?", as the answer could help generate new strategies for particle identification. However, we more directly address the question "What information can the machine be using?" which is a slightly narrower question, although we hope that the answer to the latter may help inform the answer to the former.

In this work, we set out to compare the performance of the $n$-subjettiness approach to the jet image approach for machine learned top quark tagging at three different jet $p_T$ ranges, roughly corresponding to the mostly resolved, mixed, and highly boosted regions.

This paper is structured as follows: in Section 2 we describe the details of the event generation and machine learning implementations of our heavy object taggers. In Section 3 we present the results our of top quark tagging study. We then discuss the results and conclude in Section 4.

# 2 Sample Generation and Machine Learning Techniques

## 2.1 Signal and background sample generation

In each case, we generate matrix elements at leading order (LO) in QCD and electroweak couplings with MADGRAPH5_AMC@NLO v2.60 [28] with the NNPDF 2.3 PDF set [29] interfaced via LHAPDF [30]. Idealised signal and background samples are chosen to ensure each has very little hadronic activity outside of the signal and background jets.

For our top quark tagging study we use $pp \rightarrow W_1^- t, t \rightarrow W_2^+ b, W_2^+ \rightarrow jj, W_1^- \rightarrow e^- \bar{\nu}_e$ as our signal process, and $pp \rightarrow W^- j, W \rightarrow e^- \bar{\nu}_e$ as our background. We examine three different $p_T$ ranges to capture differences in signal and background kinematics as the three-pronged top quark decay becomes more collinear: $[350, 400]$ GeV, $[500, 550]$ GeV and $[1300, 1400]$ GeV, with these cuts enforced similarly at the parton-level.

To facilitate reproduction of our results we provide the gluon to light quark ratios of our background samples in Table 1.

Table 1: Gluon to light quark ratios in the background samples.

| Background Sample | $g/q$ |
|---|---|
| $W j, p_T \in [350 - 400]$ GeV | 0.178 |
| $W j, p_T \in [500 - 550]$ GeV | 0.196 |
| $W j, p_T \in [1300 - 1400]$ GeV | 0.289 |

The Les Houches events [31] are matched to the parton shower PYTHIA 8.2.26 [32, 33] to obtain hadron-level events in the HEPMC format [34]. The input datasets for each NN are

then extracted from these by a tailored RIVET [35] analysis.

We cluster jets therein with the anti-$k_T$ algorithm throughout [36] and using the FASTJET library [37]. In our top-tagging study we cluster jets into two groups, requiring $R = 1.5$ "fat" jets within $|\eta| < 1$ in the mild and moderately boosted $p_T$ ranges $[350, 400]$ and $[500, 550]$ GeV, and standard $R = 0.8$ jets within $|\eta| < 2.5$ for the ultra-boosted top jets with $p_T$ in the range $[1300, 1400]$ GeV.

From each sample we extract from the leading $p_T$ jet both a jet image and a set of $N$-subjettiness variables as defined in Sec 2.2 and Sec 2.4 respectively, which serve as the raw pseudodata to be fed to the classifiers after further preprocessing.

These datasets consist of approximately 2M jets each divided evenly between signal and background, of which 100k are used for validation and 200k for testing.

## 2.2  Jet image generation

In order to facilitate the use with the CNNs detailed below the jets are first converted into images. Each jet is turned into a 51×51 pixel (37×37 for 1300-1400 GeV) jet image where energy deposits in a 'calorimeter' (here approximated by summing up the $p_T$ inside each pixel) are treated as the pixel intensity. The pixels span the $\eta - \phi$ space covering the entire region of the jet with a particular radius.

The particular choice of size and resolution for the jet images ensures that no information is lost while cropping the image boundaries and leads to an improvement in the accuracy relative to the initial 33×33 pixel size (as used in [13]). For the "fat" jets in the $p_T$ ranges $[350, 400]$ and $[500, 550]$ GeV, each pixel corresponds to $\Delta\eta = \Delta\phi = 0.059$. The different image size (37×37) for $R = 0.8$ jets in $p_T$ range $[1300, 1400]$ GeV follows [15] and corresponds to a slightly smaller pixel size (here, $\Delta\eta = \Delta\phi = 0.043$) and a reduced field of view which captures the details of these more narrow jets.

We implement two CNNs (CNN and CNN1) with two different architectures. The images generated undergo a series of preprocessing steps before being fed to these two CNNs. The CNN closely follows the procedure set out in [13] which are briefly described here. The $p_T$ weighted centroid of the jet is brought to the center of the jet image. This is equivalent to subtracting the $p_T$ weighted mean of $\eta$ and $\phi$, which translates the centroid to $(\eta, \phi) = (0, 0)$. The image is then cropped to 51×51 pixels in the range $\eta, \phi \in [-R, R]$. Finally, each jet image is normalised so that the summed intensity of all pixels is 1.

CNN1 follows the preprocessing steps presented in [15]. It is a modification to the DeepTop tagger [14]. The $p_T$ weighted centroid of the jet is shifted to the center of the image followed by a rotation which makes the $p_T$ weighted principal axis vertical. Images are then flipped along L-R and U-D axes. The jet image is then normalised and finally pixelated.

The application of preprocessing steps to jet images makes a CNN more robust in discriminating signal from background. Even though preprocessing steps are highly useful from a machine learning perspective, they can smear out useful information like jet mass from the image. Translation of pixels along $\eta$ can change the pixel intensity if we are considering jet energy instead of transverse momentum (transverse energy) [14]. Rotating the images does not preserve jet mass as exemplified in [12] (see page 5). Normalization of pixel intensities is useful when there are large variations in the $p_T$ of the constituents of the jet, though this operation also removes jet mass information which we then provide seperately [12, 14]. Another possible method to reduce information loss is to divide the pixel intensities by a constant instead of normalizing. This was beyond the scope of the current study but should be investigated in the future. [2]

---

[2]Thank you for the anonymous referee who suggested this.

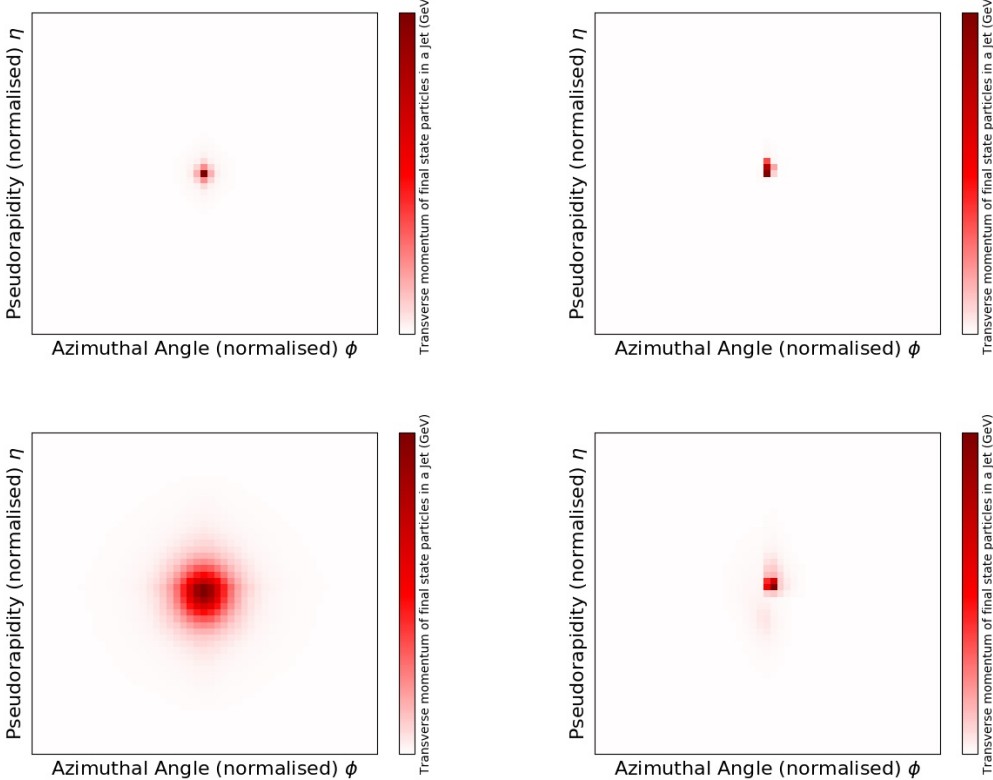

Figure 1: Average top quark signal (background) sample image on the top (bottom) after the preprocessing in [13] on the left and after preprocessing in [14] on the right.

We use only grayscale images without any additional information in the colour channels. We actually put quite a lot of effort into experimenting with including additional information like charge particle multiplicity, transverse momentum of charged particles and transverse momentum of neutral particles into the colour channels but we found that this did not give an improvement in the performance of the CNN. Jet $p_T$ and particle multiplicity were included as additional channels for the CNN implemented in [25] and this analysis gave rise to a similar performance as the grayscale images. The average images of the top quark signal and background samples of the two different preprocessing steps are shown in Figure 1.

## 2.3 Analysing jet images with Convolutional Neural Networks

Convolutional neural networks have been used in the phenomenological literature for quark/gluon classification [13], by ATLAS [38] and CMS [39], and for top quark [11, 14] and $W$-boson tagging [10, 12, 40]. The architecture of the CNNs used here are identical to Refs. [13, 25] and Ref. [14]. In both networks, jet mass is added as an additional information.

The first network (CNN) contains three convolutional layers and two fully connected layers, where He-uniform weights are used for the initialisation. The full architecture is presented in Figure 2. ReLu is used as the activation function except at the Softmax activated output [41, 42]. A batch size of 100 is used with a learning rate ($\alpha$) of 0.001. A pixel dropout of 0.1 is applied to each CNN layer.

The CNN1 network has two blocks. Each block contains two convolutional layers. Maxpooling of stride (2x2) is given at the end of each block. Three fully connected layers are also

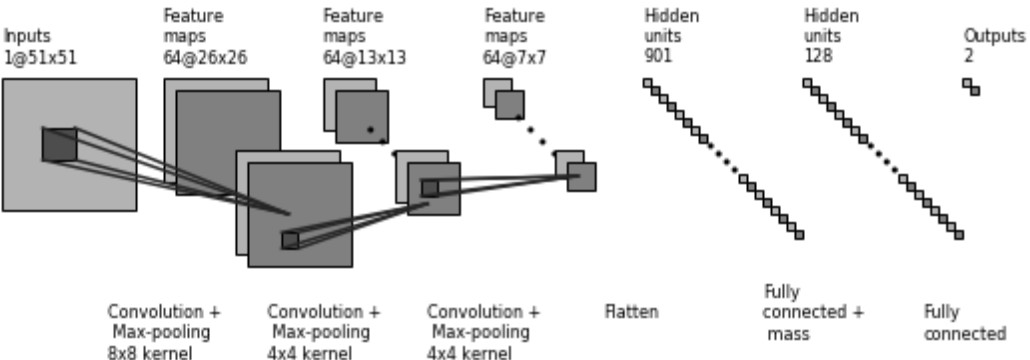

Figure 2: Network architecture of the CNN. This figure was generated by adapting the code from https://github.com/gwding/draw_convnet.

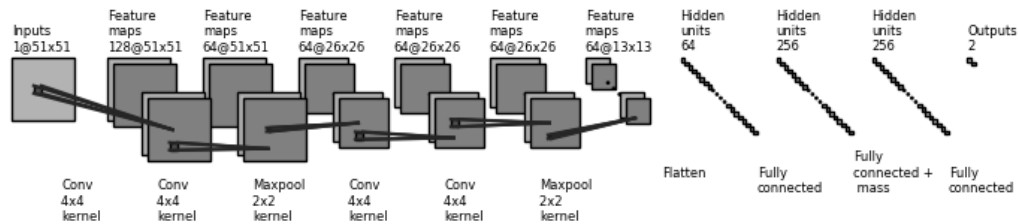

Figure 3: Network architecture of the CNN1. This figure was generated by adapting the code from https://github.com/gwding/draw_convnet.

attached to the two blocks of convolutional layers. Weights and activation functions are the same as in the first network (CNN). The network is trained using AdaDelta optimizer with a learning rate of 0.3 and without any dropout. The architecture is given in Figure 3.

The CNNs are trained for 50 epochs. Jet mass is given to the fully connected layer of the CNNs which improves the performance for all of our samples, as the preprocessing steps remove it from the input image (as explained in the pre-processing section).

TENSORFLOW 1.3.0 [43] is used to build the networks. An NVIDIA GEFORCE GTX 1080TI with CUDA 9.0 [44] is used for training.

## 2.4 Using $n$-subjettiness variables to learn the $m$-body kinematic phase space using Dense Neural Networks

Our alternative heavy object tagger is based on the proposal in [26] to use a basis of $n$-subjettiness variables [17] spanning an $m$-body phase space to teach a Dense Neural Network to separate signal and background using a relatively small set of input variables. The same idea was later used in [45] to build a tagger intended to generically distinguish QCD jets from heavy object jets, and to construct new simple observables with improved discrimination power in [27]. It was also used in a similar manner as here as a benchmark for comparing different ML implementations of heavy object tagging in [25]: the difference to the present study is that we here focus specifically on top tagging and investigate the effect of adding jet

mass information in two different implementations. The input variables are given by:

$$\left\{\tau_1^{(0.5)}, \tau_1^{(1)}, \tau_1^{(2)}, \tau_2^{(0.5)}, \tau_2^{(1)}, \tau_2^{(2)}, \ldots, \tau_{M-2}^{(0.5)}, \tau_{M-2}^{(1)}, \tau_{M-2}^{(2)}, \tau_{M-1}^{(1)}, \tau_{M-1}^{(2)}\right\},\tag{1}$$

where

$$\tau_N^{(\beta)} = \frac{1}{p_{T,J}} \sum_{i \in J} p_{T,i} \min\left\{R_{1i}^{\beta}, R_{2i}^{\beta}, \ldots, R_{Ni}^{\beta}\right\}\tag{2}$$

and $R_{ni}$ is the distance in the $\eta - \phi$ plane of the jet constituent $i$ to the axis $n$. We choose the $N$ axes using the $k_T$ algorithm [46] with $E$-scheme recombination [47][3].

There are several benefits to this choice of input variables: for one, it allows an intuitive understanding of *what the network actually learns* since enlarging the input data set corresponds to allowing the network to access higher body kinematic phase space information, and the point where the network saturates can therefore be given a physical interpretation.

Additionally, $n$-subjettiness variables are very well-understood and are some of the most studied and used jet substructure observables with for example ATLAS detector-level and unfolded distributions for $\tau_2^{(1)}/\tau_1^{(1)}$ and $\tau_3^{(1)}/\tau_2^{(1)}$ in a dijet sample having been published in [20].

We train simple DNNs using these as input nodes. They consists of four fully connected hidden layers, the first two with 300 nodes and a dropout regularisation [48] of 0.2, and the last two with 100 nodes and a dropout regularisation of 0.1. The output layer consists of two nodes. We use the ReLu activation function [41] throughout and minimise the cross-entropy using Adam optimisation [49]. To add jet mass information we simply include it among the input parameters in Equation 1.

We implement this DNN using CUDA 9.0 [44] and TENSORFLOW 1.8.0 [43]. An NVIDIA GEFORCE GTX 1080 is used for training. The number of epochs trained depends on the input data used and ranges from 500-1500 to ensure convergence.

## 3 Top Quark Tagging Results

The receiver operating characteristic (ROC) curves for the image networks and the $n$-subjettiness networks with up to 8-body phase space information (to show that the information saturates at 4-body phase space) are presented in Figures 4, 5, and 6, with results without mass information on the left and with mass information on the right. The area-under-curve values for the ROC curves for a selection of network configurations are presented in Table 2. In general the performance of the two methods is very comparable both with and without jet mass information included, which suggests the image networks ultimately are probing very similar information as the $n$-subjettiness one. This is a non-trivial test of whether or not our image network (as we have implemented it here) accesses information which can not be considered safe from a modeling perspective: since it saturates at 4-body kinematic phase space information, we can be fairly certain it is learning features of the hard splittings and first few splittings in the parton shower rather than low-energy features which should not be trusted.

The similar performance of the image network and saturated $n$-subjettiness network without mass information and the large and comparable improvements that can be seen by including the mass information (which is itself also included in the plot with mass information) can be understood by considering the information included in the network inputs without mass: in the CNN case, the pre-processing steps we take smear out the mass information considerably [12,14], and in the $n$-subjettiness once we effectively remove all jet scales from the input

---

[3]This choice is identical to that in [26] and we have not attempted to optimise it any further, since (as we will show) we get very competitive performance 'out of the box'.

Table 2: The area-under-curve (AUC) values for a selection of our ROC curves. Larger values are better and AUC=1 corresponds to perfect signal and background discrimination.

| Sample | mass + CNN1 | mass + 3-body | mass + 5-body |
|---|---|---|---|
| Top $p_T \in [350 - 400]$ GeV | 0.9626 | 0.9503 | 0.9613 |
| Top $p_T \in [500 - 550]$ GeV | 0.9678 | 0.9535 | 0.9658 |
| Top $p_T \in [1300 - 1400]$ GeV | 0.9698 | 0.9607 | 0.9723 |

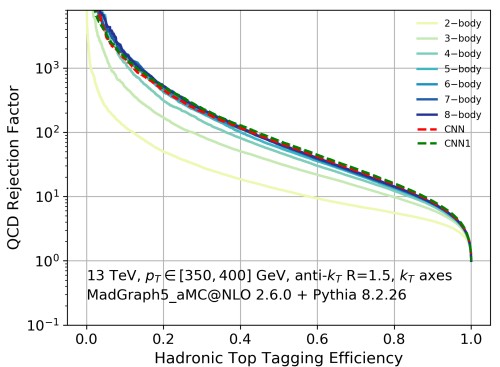 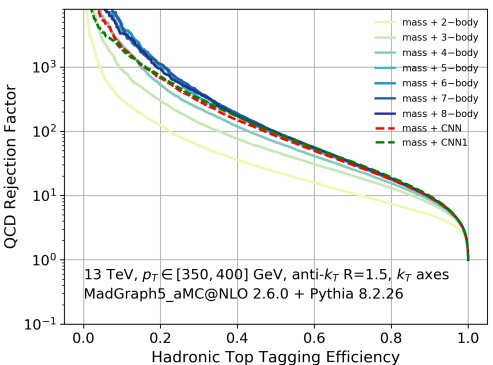

Figure 4: ROC curves for top quark tagging without mass on the left and with mass on the right, for $p_T \in [350, 400]$ GeV. Adding mass information improves the performance of the image networks and the $n$-subjettiness network.

information by using the normalised definition of $\tau_N^{(\beta)}$ in Equation 2, smearing any jet mass information the network can deduce from the $n$-subjettiness variables themselves.

This suggests only the results with mass information included should be considered when answering the question of what the image networks are learning with reference to the $n$-subjettiness results. The image network therefore shows excellent agreement with the saturated $n$-subjettiness network, which this time happens at 8-body kinematic phase space for $p_T \in [350, 400]$ GeV and $p_T \in [500, 550]$ GeV and 5-body for $p_T \in [1300, 1400]$ GeV. The $p_T$ dependence of the point where the networks saturate should not be oversold, however: the differences are small and could potentially be removed by improving the network architecture and hyperparameter optimisation.

We also plotted pairwise correlations for a subset of the dataset. Figures 7-9 represent how $n$-subjettiness variables in different $m$-body phase space correlate within themselves as well as with the image networks. In these images the horizonal and vertical images represent the output probability of the network (one for top quark, zero for background) and the shading represents how many events fell in each pixel, there being around $10^5$ events for each case.

## 4 Discussion and Conclusion

From our results it is clear that the performance of a CNN image networks at heavy object tagging can be closely matched by a simpler DNN using $n$-subjettiness variables for top quarks at a number of $p_T$ ranges, once mass information is added to both networks in a consistent manner. This clarifies the question of what the image networks are actually learning, as the

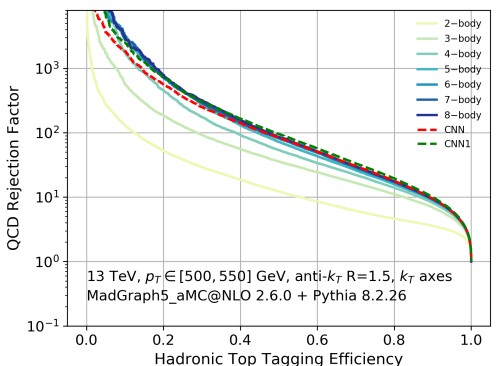 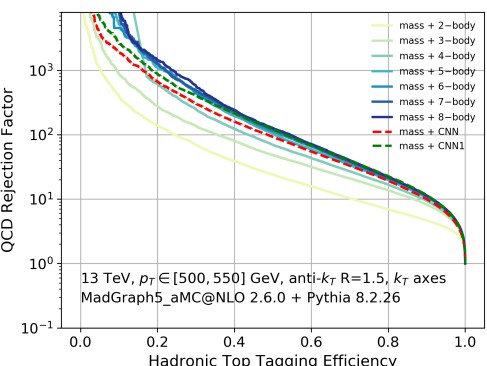

Figure 5: ROC curves for top quark tagging without mass on the left and with mass on the right, for $p_T \in [500, 550]$ GeV. In this case the performance after adding mass information is very similar.

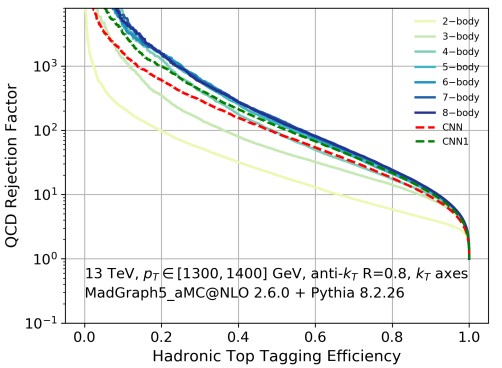 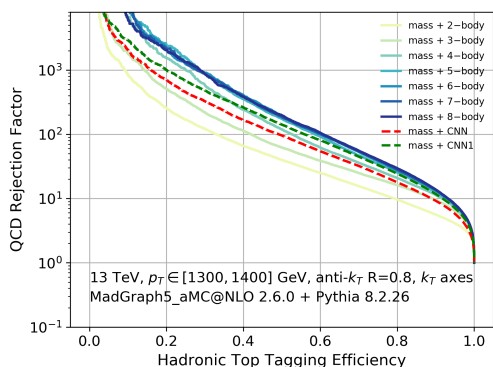

Figure 6: ROC curves for top quark tagging without mass on the left and with mass on the right, for $p_T \in [1300, 1400]$ GeV. The image networks once again under-perform before adding mass information, but is slightly better at high signal efficiency after mass information is added.

saturated $n$-subjettiness network gives an intuitive mapping to $m$-body kinematic phase space which suggests the information accessed is mostly contained in 4- and 5-body (and in the case of highly boosted top quarks, 8-body) kinematic phase space, which can be expected to be well-modeled by modern Monte Carlo event generators. The small differences that remain are better explained by differences in hyperparameter optimisation and overall architecture of the networks than differences in the underlying information accessed.

It is interesting to note that [25] found comparable performance between jet image networks, $n$-subjettiness networks, and a linear ML algorithm making use of EFPs at a number of different tagging applications as well. Due to the simpler structure and training of linear ML algorithms and EFPs forming a complete linear basis of jet substructure observables, this suggests all of these methods are doing a good job at accessing the full substructure information available in a generic jet.[4] This conclusion is also supported by the performance of the ML tagger operating directly on the kinematic information of individual final state particles presented in [50]: when taking detector granularity into account, such information only becomes

---

[4]We note that we find slightly better agreement between the jet image and $n$-subjettiness results than [25], especially for top quark tagging, however this could be due to the manner in which we include the jet mass information.

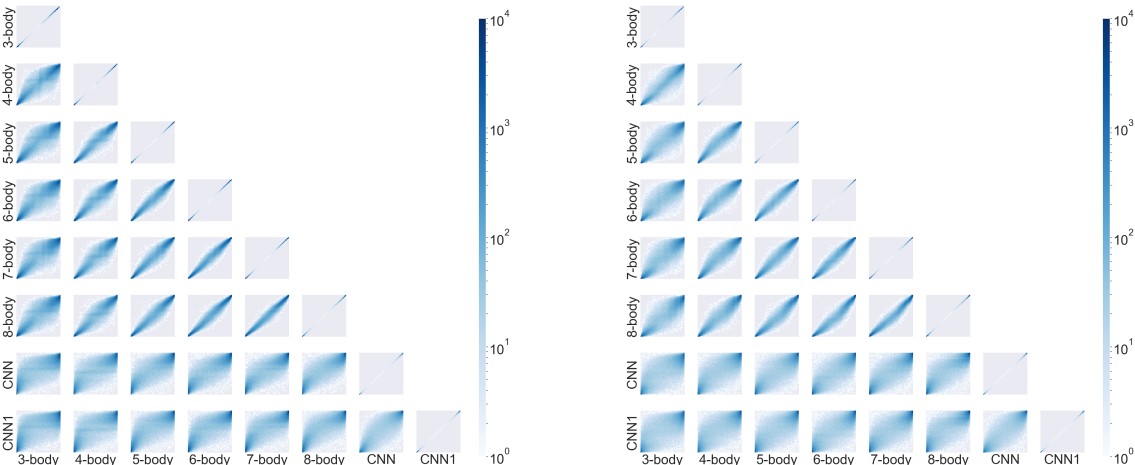

Figure 7: Correlation plots for top quark tagging without mass on the left and with mass on the right, for $p_T \in [350, 400]$ GeV. There are strong correlations between $n$-subjettiness variables in different $m$-body phase space. Adding mass to the network increases the spread of the correlation in the image networks.

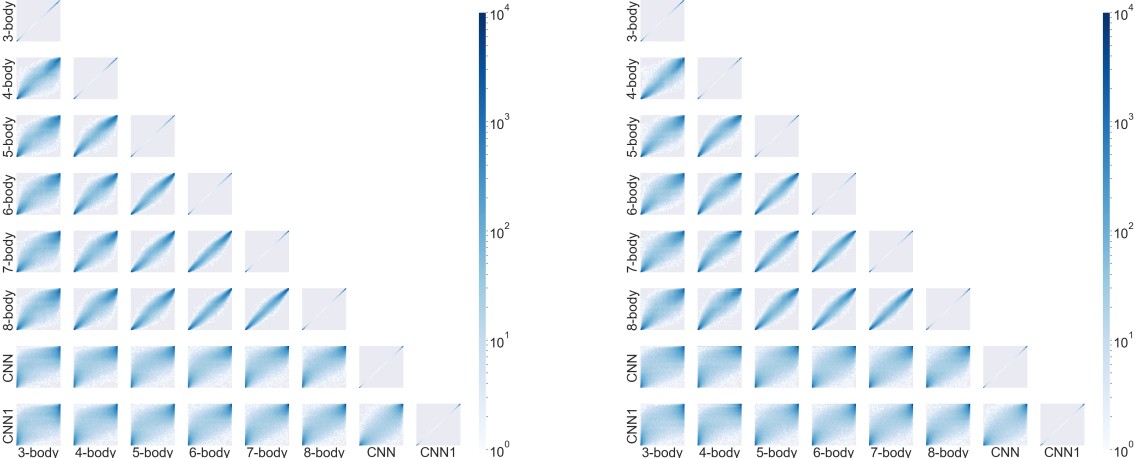

Figure 8: Correlation plots for top quark tagging without mass on the left and with mass on the right, for $p_T \in [500, 550]$ GeV.

necessary at very high boosts where the calorimeters are too granular to resolve the hard decay, but the ultimate performance of the tagger matches a comparable image networks to the one presented here (with mass information taken into account by only taking pre-processing steps which don't smear it out) for top quarks with $p_T \in [350, 400]$ GeV. We note that the $n$-subjettiness network presented here also can take particle flow information into account, and therefore should remain competitive at large boosts.

To express these conclusions in a different way, it appears that infrared safe information is enough to saturate the performance of tagging algorithms of multi-prong topologies from resonant decays, even when comparing to image methods which explicitly include non-infrared safe information. Our result should generalise to any observable basis which contains the full infrared safe information up to some M-body order, which could potentially be constructed from observables that are simpler to calculate than n-subjettiness.

Having established that heavy object tagging using jet image and $n$-subjettiness variables offer comparable performance in an idealised setting, it is interesting to also ask the question of to what extent this agreement survives under a more realistic setting where there are additional

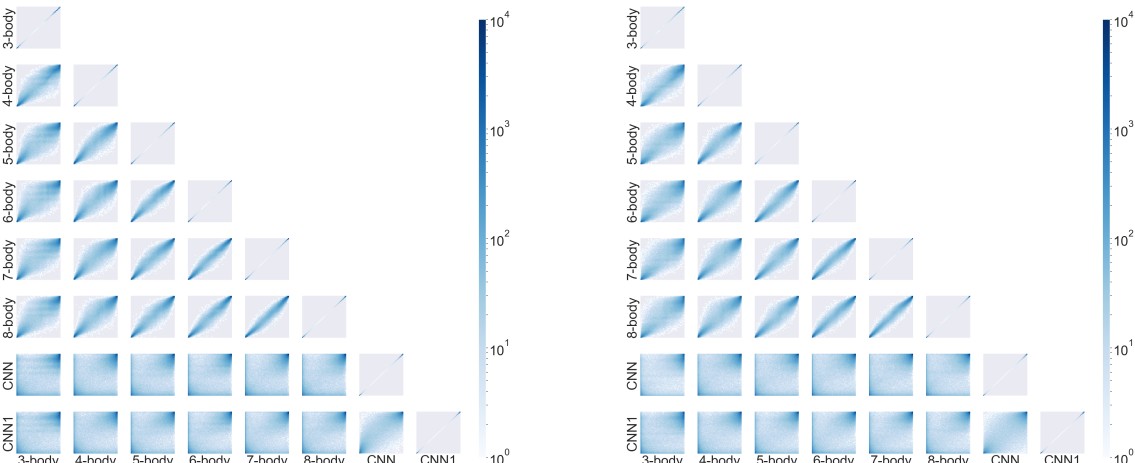

Figure 9: Correlation plots for top quark tagging without mass on the left and with mass on the right, for $p_T \in [1300, 1400]$ GeV. The weak correlation corresponds to the poor performance of the image networks in the ROC curves in Figure 6

coloured particles and pile-up in the final state, background samples are more complex, and detector effects are taken into account. We leave such a study to future work.

# Acknowledgements

We would like to thank Kalliopi Petraki, Andy Buckley, Michael Russell, and Anja Butter for useful discussions. SV and MF would like to thank Robert Hogan for many related discussions and support. The work of MF was supported partly by the STFC Grant ST/L000326/1. SV and MF are also funded by the European Research Council under the European Union's Horizon 2020 programme (ERC Grant Agreement no.648680 DARKHORIZONS). SV was the recipient of a Rick Trainor PhD Scholarship at the start of her studies. KN is supported by the NWO Vidi grant "Self-interacting asymmetric dark matter". LM is supported by the FNRS under the IISN convention "Maximising the LHC physics potential by advanced simulation tools and data analysis methods" .

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
