# Peer review of "Reports of My Demise Are Greatly Exaggerated: $N$-subjettiness Taggers Take On Jet Images"

_SciPost Physics, doi:SciPost Phys. 7, 036 (2019)_

## Round 2 · Referee Report · Anonymous · 2019-6-27

Strengths
1. A robust comparison of different representations of jets for machine learning is provided.
2. The type(s) of information used by the machine learning methods is probed.
3. The importance of mass smearing issues with standard pre-processing techniques is clearly discussed.
4. The writing in the paper is clear and understandable.
Weaknesses
1. The comparison is restricted to calorimeter/infrared-safe information, without significant discussion of how additional information (counts, charge, flavor) can change the conclusions.
2. Important experimental issues such as pileup and detector effects are not included.
3. A more detailed discussion of the physics of different mass smearing effects is lacking.
Report
This paper provides a direct comparisons of two very different feature representations of jets for machine learning applications: jet-images with convolutional neural networks (CNNs) and a basis of N-subjettiness observables with dense neural networks (DNNs). The two jet representations are found to provide consistent performance, using the identification of hadronic top decays in semi- and highly-boosted limits as a case study. Further, the relevance of jet mass information (smeared by pre-processing steps) for tagging resonant decays is highlighted quantitatively.
This paper tackles the important question of how to explore different jet representations and what information the machine is using. These issues will become increasingly important as these powerful modern machine learning tools advance toward use in experimental analyses. I believe that this paper will be suitable for publication in SciPost once the questions and points discussed below are suitably addressed.
As the studies in the paper appear robust and technically sound, I think it is only necessary to add discussions and clarifications in response to the questions and comments below.
Questions and comments:
1. What determined the pixelization choices (51x51 and 37x37)? Are these intended to reflect experimental resolutions, to optimize the training of the CNN, or something else?
2. Translating the maximum-pT pixel to (eta,phi)=(0,0) is not equivalent to subtracting the pT-weighted centroid, i.e. translating the centroid to (eta,phi)=(0,0). Both of these are sensible pre-processing choices, but they are not the same (as claimed in the paper) since the max pixel and the pT-mean pixel may be different. It is worth clarifying this in the text.
3. How do the conclusions change with the introduction of additional information (particle types, counts, charges, etc)? How can a similar comparison study be done, even in principle? The jet images can include multiple colour channels to accomodate this information, whereas the N-subjettiness observables cannot easily do this.
4. The importance of jet mass is a crucial element of this paper. There are different pre-processing steps that smear the jet mass to different extents, which would be important to enumerate in the paper. For instance, normalizing the pT to 1 impacts both the jet images and the N-subjettiness observables. Discretization affects both CNN and CNN1. Rotation affects only CNN1.
5. If the pT were divided by a constant (e.g. pT->pT/400 GeV for the low pT sample) instead of being normalized to sum to 1, it would provide some of the preprocessing benefits without significantly smearing the mass. Would the authors advocate for something like this instead of adding the mass value back in separately?
6. What are the scope of your conclusions about observable bases vs. image representations and the relevance of mass? It is worth spelling them out more clearly in the conclusions. To me, it seems that conclusions appear generally relevant for using infrared safe information to identify multi-prong topologies originating from resonant decays.
7. The question of "What is the machine learning?" is a broad one of great recent interest with these black-box methods. This paper appears to probe the slightly more narrow question of "What information can the machine be using?" by restricting to N-body kinematics. This is a semantic issue, but it may be worth clarifying in the introduction that this is specifically what will be explored in the paper.
Small typos or wording issues:
1. p. 4: "shown in 1." should be "shown in Figure 1."
2. p. 6: "50 epochs . As". There is a period issue here, and the next sentence appears to be a non sequitur.
3. p. 7: "The output layer are two nodes" sounds a little odd. Perhaps "consists of two nodes" would be more suitable?
4. Fig. 6: "The image networks once again under-performs" would be better as "under-perform".
5. p. 8: "Figures ??" should be something like "Figures 7-9".
Requested changes
Requested changes:
1. Discuss the choice of pixelization and revise the discussion of translations in Sec. 2.2.
2. Include additional discussion of the mass smearing for the different methods.
3. Add discussion about how additional information (particle-type, charge, counts, etc.) would affect the conclusions.
4. Spell out the scope and impact of the conclusions more clearly in Sec. 4.
5. Clarify the aspect(s) of "what the machine is learning" that will be probed in the paper.
6. In the introduction, include some relevant references to the alluded-to experimental uses of refs. [2-9].
Author: Sreedevi Varma on 2019-07-16 [id 564]
(in reply to Report 1 on 2019-06-27)This is the first time we have used scipost and we don't understand the procedure. We have an updated manuscript with the changes below, but we don't know where to put it. We'll put the reply here but where do we put the updated manuscript??? Anyway, here is the normal reply:-
Thank you very much for your careful report which will improve the paper. Here
we outline the corrections we have made in order to address your comments.
Q. What determined the pixelization choices (51x51 and 37x37)? Are these
intended to reflect experimental resolutions, to optimize the training of the
CNN, or something else?
A. We used these pixel sizes as it gave an improvement in the performance
of the network. We have included a paragraph in section 2.2 elaborating
this.
Q. Translating the maximum-pT pixel to (eta,phi)=(0,0) is not equivalent
to subtracting the pT-weighted centroid, i.e. translating the centroid to
(eta,phi)=(0,0). Both of these are sensible pre-processing choices, but they
are not the same (as claimed in the paper) since the max pixel and the pT-
mean pixel may be different. It is worth clarifying this in the text.
A. This was a mistake we made in the wording of the paper which did not ex-
tend to our calculations or numerics. We are translating the pT-weighted
centroid to the center of the image for both CNN and CNN1. We have
corrected the sentence, hopefully it is more accurate and clear now.
Q. How do the conclusions change with the introduction of additional infor-
mation (particle types, counts, charges, etc)? How can a similar compari-
son study be done, even in principle? The jet images can include multiple
colour channels to accomodate this information, whereas the N-subjettiness
observables cannot easily do this.
A. We have tried to add colour channels but it didn’t give any improvement
to the performance, we have now spelled this out clearly in the paper.
Q. The importance of jet mass is a crucial element of this paper. There are
different pre-processing steps that smear the jet mass to different extents,
which would be important to enumerate in the paper. For instance, nor-
malizing the pT to 1 impacts both the jet images and the N-subjettiness observables. Discretization affects both CNN and CNN1. Rotation affects only CNN1.
A. We have added a brief description of how information is lost in prepro-
cessing steps in section 2.2
Q. If the pT were divided by a constant (e.g. pT →pT/400 GeV for the low pT
sample) instead of being normalized to sum to 1, it would provide some of
the preprocessing benefits without significantly smearing the mass. Would
the authors advocate for something like this instead of adding the mass
value back in separately?
A. Thank you for making this point. We haven’t considered this in our cur-
rent study. We’ve commented on it in the new version.
Q. What are the scope of your conclusions about observable bases vs. image
representations and the relevance of mass? It is worth spelling them out
more clearly in the conclusions. To me, it seems that conclusions appear
generally relevant for using infrared safe information to identify multi-
prong topologies originating from resonant decays.
A. we added a paragraph to our conclusions which we think addresses this
question.
Q. The question of “What is the machine learning?” is a broad one of great
recent interest with these black-box methods. This paper appears to probe
the slightly more narrow question of “What information can the machine
be using?” by restricting to N-body kinematics. This is a semantic issue,
but it may be worth clarifying in the introduction that this is specifically
what will be explored in the paper.
A. we have added a new paragraph in the introduction, (which borrows heav-
ily from your comment here, hope you don’t mind) we agree it is worth
pointing this out a bit more clearly.
Small typos or wording issues:
1.
p. 4: “shown in 1.” should be “shown in Figure 1.”
2.
p. 6: “50 epochs . As”. There is a period issue here, and the next sentence
appears to be a non sequitur.
We have changed the paragraph to “The CNNs are trained for 50 epochs.
Jet mass is given to the fully connected layer of the CNNs which improves
the performance for all of our samples, as the preprocessing steps remove
it from the input image (as explained in the pre-processing section”
3.
p. 7: “The output layer are two nodes” sounds a little odd. Perhaps
“consists of two nodes” would be more suitable?
4.
Fig. 6: “The image networks once again under-performs” would be better
as “under-perform”.
2
5.
p. 8: “Figures ??” should be something like “Figures 7-9”.
We have corrected all of these typos.

---

## Round 3 · Referee Report · Anonymous · 2019-8-24

Report

I thank the authors for suitably addressing my comments and questions. The present version of the manuscript is much improved, and I now recommend it for publication in SciPost.

---

## Editorial Decision

published